# Fluence Dependence of Surface Morphology and Deuterium Retention in W Bulks and Nanocrystalline W Films Exposed to Deuterium Plasma

**Jing Yan** [1] , **Xia Li** [1] **and Kaigui Zhu** [1,2,*]

1    Department of Physics, Beihang University, Beijing 100191, China; jingyan@buaa.edu.cn (J.Y.); BY1619108@buaa.edu.cn (X.L.)
2    Beijing Key Laboratory of Advanced Nuclear Energy Materials and Physics, Beihang University, Beijing 100191, China
*    Correspondence: kgzhu@buaa.edu.cn; Tel.: +86-159-1097-7162

**Abstract:** The surface morphology of pure W bulks and nanocrystalline tungsten films was investigated after exposure to a low-energy (100 eV/D), high-flux ($1.8 \times 10^{21}$ D·m$^{-2}$s$^{-1}$) deuterium plasma. Nanocrystalline tungsten films of 6 μm thickness were deposited on tungsten bulks and exposed to deuterium plasma at various fluences ranging from $1.30 \times 10^{25}$ to $5.18 \times 10^{25}$ D·m$^{-2}$. Changes in surface morphology from before to after irradiation were studied with scanning electron microscopy (SEM). The W bulk exposed to low-fluence plasma ($1.30 \times 10^{25}$ D·m$^{-2}$) shows blisters. The blisters on the W bulk irradiated to higher-fluence plasma are much larger (~2 μm). The blisters on the surface of W films are smaller in size and lower in density than those of the W bulks. In addition, the modifications exhibit the appearance of cracks below the surface after deuterium plasma irradiation. It is suggested that the blisters are caused by the diffusion and aggregation of the deuterium-vacancy clusters. The deuterium retention of the W bulks and nanocrystalline tungsten films was studied using thermal desorption spectroscopy (TDS). The retention of deuterium in W bulks and W films increases with increasing deuterium plasma fluence when irradiated at 500 K.

**Keywords:** nanocrystalline tungsten films; magnetron sputtering; deuterium retention

## 1. Introduction

Tungsten (W) is being considered for use as one of the plasma-facing materials (PFMs) in future nuclear fusion devices. W is regarded as a favorable PFM due to its high melting point, low erosion rate, high thermal conductivity, etc. [1]. As a plasma-facing material, the W surface would be irradiated by magnetically confined plasma, such as helium and hydrogen isotope particles [2]. Hydrogen isotopes would be implanted and retained in the PFM, since this may affect the safe and stable operation of the fusion reactor devices. Changes in surface morphology may result in the degradation of material properties and increase erosion formation. Increased retention has a significant impact on tritium inventory in International Thermonuclear Experimental Reactor (ITER). To estimate the effect of hydrogen isotopes in future fusion reactors, it is necessary to understand the behavior of hydrogen isotopes in plasma-facing materials.

Tungsten materials have been exposed to different doses of deuterium plasmas, which allows the opportunity to predict the behavior of deuterium in tungsten [1,3–5]. Studies have shown that irradiation damage such as surface changes can strongly increase the retention of hydrogen isotopes [1,3]. The deuterium retention in polycrystalline tungsten irradiated with different ion fluences was studied, and it was reported that the retention of deuterium increases with increasing incident D ion fluence at 500 K [4]. Furthermore, the morphology changes and deuterium retention behaviors of magnetron sputtering W films are significantly different from those of bulk W, because deposited W films are considered

to be defect-rich materials. For comparison, high-energy deuterium ions were implanted into W films and polycrystalline W bulk. It was found that the surface of W films with more grain boundaries changed little after irradiation [5]. Consisting of a large volume fraction of grain boundaries and a mass of grains, nanocrystalline films usually have better properties, differing from those of the bulk materials with large grain size. Nanocrystalline film of high grain-boundary density is considered to be a high radiation-resistant material as the grain boundaries play a decisive role in irradiation resistance, which reduces the accumulation of helium inside the grains and mitigates the local supersaturation of helium [6].

Therefore, we focused on nanocrystalline materials and prepared nanocrystalline W films by magnetron sputtering deposition, continuing to study this favorable kind of materials. Deuterium retention behavior and morphology changes in nanocrystalline W films under different irradiation fluences have not been sufficiently investigated. Here we describe an experiment to determine whether irradiation fluence increases deuterium retention in tungsten exposed to fusion plasmas in a tokamak divertor environment. In this work, experiments are described using W bulks and nanocrystalline W films with different morphologies and microstructures. Nanocrystalline W films prepared by magnetron sputtering deposition were used, and deuterium ions were implanted into nanocrystalline W films by a high-flux plasma source. Meanwhile, deuterium retention in nanocrystalline W films was investigated over a wide range of implantation fluences, from $1.30 \times 10^{25}$ to $5.18 \times 10^{25}$ m$^{-2}$. W bulks were used for control experiments. The main purpose of this study is to determine morphology changes and retention behaviors in W bulks and nanocrystalline W films exposed to high-flux plasmas at different irradiation fluences and to understand their irradiation behavior.

## 2. Experimental Procedures

### 2.1. Preparation of Specimens

High-purity polycrystalline W bulk (purity: 99.95%) was purchased from Advanced Technology & Materials Co. Ltd (Beijing, China). It was made by powder metallurgy and stress-relieved by annealing at 1273 K for 1 h after hot rolling. The W bulk was 70 mm in diameter and 5 mm in thickness, and was placed horizontally in a commercial magnetron sputtering device (KYKY MP 650-A) as a target. This W bulk material was also used as the substrate for magnetron sputtering deposition. For the present experiments, the substrates were cut into $10 \times 10$ mm$^2$ pieces, each 1 mm thick. Each substrate was mechanically and electrochemically polished until the surface was mirror-like before irradiation.

Nanocrystalline W films were deposited in a magnetron system (KYKY MP 650-A) by dc sputtering of a tungsten cathode. The W films were prepared by sputtering of the W target mentioned above with 100 W dc power. The system was pumped down to a base pressure of less than $5 \times 10^{-4}$ Pa. Argon (Ar) gas was introduced via a mass flow controller. Deposition was performed in an argon atmosphere at a pressure of 1 Pa. The substrate temperature was heated up to 773 K before deposition. Tungsten films with thicknesses of about 6 μm were achieved, which were measured by a profilometer (Dektak 6 M, Veeco).

Furthermore, the W film specimens were annealed at 1273 K for 1 h in a vacuum (<10$^{-6}$ Pa) to relieve material stresses and to reduce intrinsic defect concentration. After that, specimens were electrochemically polished to remove surface impurities. Prior to each implantation, the samples were ultrasonically cleaned in acetone, alcohol, and deionized water for 15 min.

The density of the W film was measured by Archimedes' method, and it was about 17.5 g/cm$^3$ for the W film, corresponding to a relative density of 90%. The oxygen content of the W film was measured by a Nitrogen/Oxygen Analyzer (TC600, LECO, San Jose, CA, USA), and it was about 0.03 wt%. The carbon content of the film was characterized by an energy dispersive spectrometer (EDS, IE 300 X) and was about 0.05 wt%.

## 2.2. Deuterium Implantation by Plasma Exposure

All irradiation experiments were finished in an ultra-high vacuum accelerator device using deuterium ions at normal incidence to the specimens; this device is the linear plasma generator comprehensive ECR plasma for tritium (CEPT) consisting of an electron-cyclotron resonance (ECR) plasma source at the Science and Technology on Surface Physics and Chemistry Laboratory, Jiangyou, Sichuan Province. Microwaves (2.45 GHz) are coupled into the vacuum vessel through a waveguide. The magnetic field is created by four magnetic coils. CEPT can produce a steady-state plasma confined by a steady-state magnetic field of 0.25 T. A diverging plasma beam impinges perpendicularly onto the substrates. The energy of the ions impinging on the substrates is controlled by applying a dc bias to the substrate electrode. The background pressure was lower than $5 \times 10^{-5}$ Pa, and below 0.6 Pa during implantation. The specimens were bombarded by a 100 eV D beam, which was below the energy threshold for creating lattice defects by atomic displacements and causing significant damage to the surface by sputtering. The D ion flux was about $1.8 \times 10^{21}$ m$^{-2}$ s detected by a Langmuir probe (ESPION). The specimens were exposed to deuterium plasma at several different fluences: $1.30 \times 10^{25}$, $2.59 \times 10^{25}$, and $5.18 \times 10^{25}$ D/m$^2$ with the flux controlled at $1.8 \times 10^{21}$ D/m$^2$ s. The required fluences were achieved by adjusting the duration of the exposure (2, 4, and 8 h). During deuterium plasma bombardment, the specimen's temperature was kept at 500 K, controlled by an external open-circuit thermostat using cooling water. After the irradiation, the ion beam was turned off and the temperature of specimens was allowed cool to room temperature. The low fluence and high fluence were primarily controlled by the exposure time; thus, the specimens were exposed to plasma at different irradiation durations. The detailed experimental parameters are represented in Table 1. The difference between each set of experiments was that only one parameter changed, while all the other parameters remained as close as possible. By changing the exposure plasma fluences (exposure duration), we were able to investigate its effects on surface morphology changes and deuterium retention.

**Table 1.** Overview of the exposure conditions of the three investigated samples.

|  | W Bulk | W Film | W Bulk | W Film | W Bulk | W Film |
|---|---|---|---|---|---|---|
| **Flux (m$^2$s)** | $1.8 \times 10^{21}$ | | $1.8 \times 10^{21}$ | | $1.8 \times 10^{21}$ | |
| **Duration (h)** | 2 | | 4 | | 8 | |
| **Fluence (m$^2$)** | $1.30 \times 10^{25}$ | | $2.59 \times 10^{25}$ | | $5.18 \times 10^{25}$ | |
| **Temperature (K)** | 500 | | 500 | | 500 | |
| **Ion energy (eV)** | 100 | | 100 | | 100 | |

## 2.3. Sample Characterization

The surface topography was analyzed by field-emission scanning electron microscopy (SEM) (JSM-6701F; JEOL, Tokyo, Japan). SEM was used to analyze the surface morphology changes of deposited W films from before to after deuterium implantation, and the samples were inclined by 45 degrees. The internal features of blisters below the sample surface were investigated using a scanning electron microscope (SEM) equipped with a focused ion beam (FIB) for the cross-sections of blisters. Thermal desorption spectroscopy (TDS) was utilized to obtain information about the binding energies of deuterium and release behaviors in tungsten films after deuterium plasma exposure. The total amount of deuterium retained in the tungsten films after implantation was also measured by means of TDS. In this paper, the spectra of the thermal desorption spectrum of deuterium released from the samples with temperature are given. In all TDS measurements, the maximum temperature is set at 1273 K with a linear temperature ramping rate of 10 K/min, which was monitored by a K-type thermocouple. The amount of D retained in the specimens was determined by integrating the quadrupole mass spectrometer (QMS) signals for molecules D2, HD (signals of HDO and D2O were negligibly small) from the start of heating ramp-up to the time when the maximum temperature was reached. The number of desorption gases was calculated by integrating the time of the desorption rate.

## 3. Results

### 3.1. X-ray Diffraction Patterns

Figure 1 displays the X-ray diffraction patterns of the W bulks and W films before and after irradiation. The W bulk and W film both have bcc structure and present four diffraction peaks, located at 40.0°, 58.0°, 73.0°, and 87.0°, which correspond to the (110), (200), (211), and (220) crystal orientation, respectively. No obvious preferred orientation is shown in the patterns, compared with the standard card (PDF#04-0806: tungsten). In addition, the full width at half-maximum (FWHM) of W film is higher than that of W bulk, indicating the grain size of W film is smaller compared with that of W bulk. After irradiation, the peak positions of both the W bulk and W film shift slightly to lower diffraction angles as the deuterium fluence increases (see Figure 1a–d). By observing the (110) diffraction peaks, it can be seen that the (110) diffraction peaks of both the W bulks and W films shift to smaller 2θ angle (see Figure 1e–h).

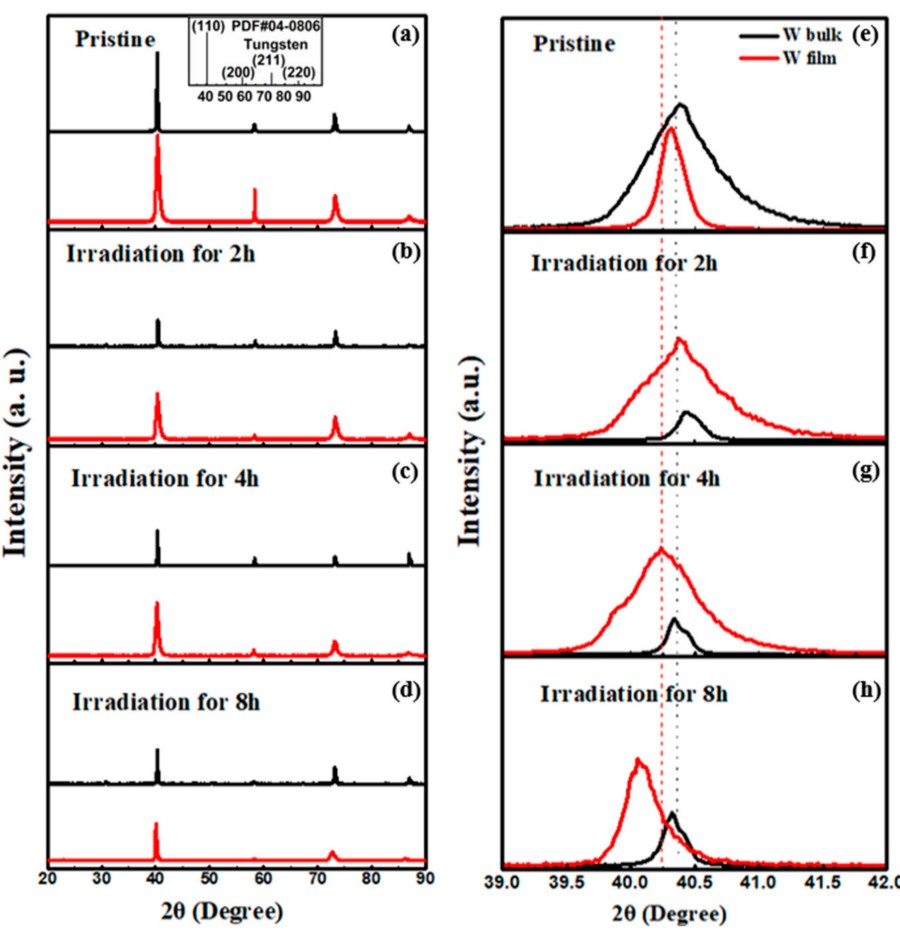

**Figure 1.** Pre- and post-deuterium irradiation X-ray diffraction (XRD) scans for W bulks and nanocrystalline W films.

### 3.2. Nanoindentation Analysis

The hardness of the W bulk and W film before irradiation was investigated, as seen in Figure 2. The nanoindentation test provided information on the comparison of the two samples. To obtain the hardness value, we set the maximum indentation depth at 1 μm and the indentation strain rate at about 0.05 s$^{-1}$. The tip radius of the indenter was estimated to be 160 nm. The hardness of W film is 17.5 GPa and of W bulk is 7.6 GPa. The hardness value of pure W film is higher than that of W bulk. These nanoindentation results indicate that the hardness of W is improved by the preparation of nanocrystalline W films by magnetron sputtering.

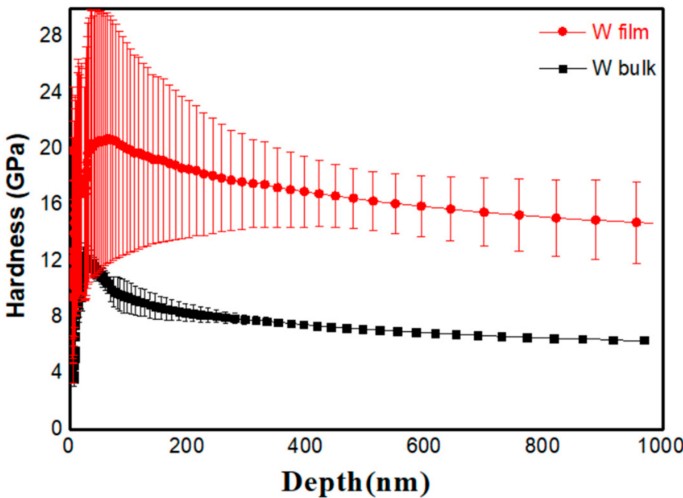

**Figure 2.** Hardness displacement curves for pure W bulk and W film before exposure.

### 3.3. Blister Formation

Figure 3a,b shows surface images of the pure W bulk and magnetron sputtering nanocrystalline W film before irradiation, respectively. For the pure W bulk, the typical grain sizes are between 10 and 50 μm. The grain sizes of W film were measured through the electron channeling contrast technique in scanning electron microscopy, and it is clearly seen that the average grain size of W film is approximately 100 nm.

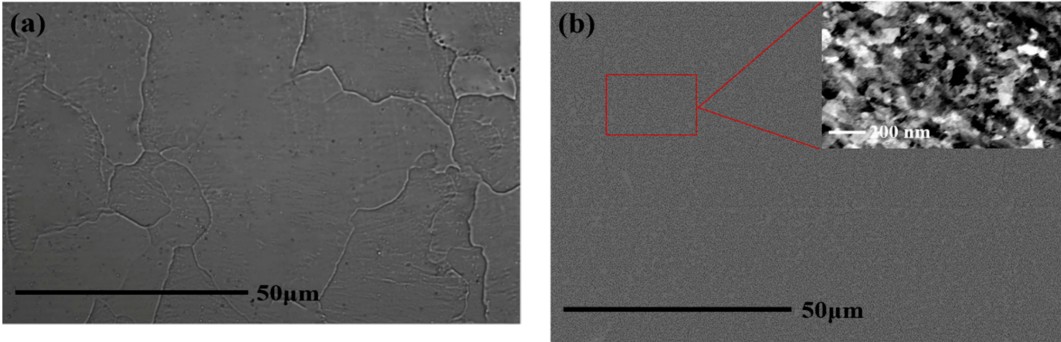

**Figure 3.** Pre-irradiation scanning electron microscopy (SEM) images for (**a**) pure W bulk and (**b**) nanocrystalline W film samples.

The top-view SEM images of the surface morphology of the W bulks and W films exposed to D plasma at different fluences are shown in Figure 4a–f. The surface morphology differs significantly, and blisters were observed on the samples exposed at a surface temperature of 500 K and ion energy of 100 eV, as shown in Figure 4a–f. A few micrometer-sized blisters are observed (Figure 4) on the surface of samples inclined by 45 degrees. The size of blisters is less than a few micrometers. It has been reported that blisters are observed on W surface after D plasma irradiation at ~500 K [7,8]. The W bulk exposed to low-fluence plasma ($1.30 \times 10^{25}$ D·m$^{-2}$) shows blisters about 500 μm in diameter (Figure 4a). The blisters on samples irradiated to higher-fluence plasma are much larger (~2 μm), as seen in Figure 4e. Compared with those of W bulk, the size and density of blisters on the surface of W film are smaller and lower, respectively. Analyzing images taken with a 45 degree tilt angle, the shape of blisters is close to elliptic. The blisters show a multilayer step-like structure, which is similar to those described in tungsten exposed to deuterium plasma or low-energy ion implantation in Ref. [8]. The influence of the plasma fluences on the blistering behavior is examined for fluences in the range of $1.30 \times 10^{25}$ to $5.18 \times 10^{25}$ D/m$^2$, while the other conditions are held constant. With the ion fluence in-

crease, the size of blisters increases both on the W bulks and on the W films. It is speculated that the nucleation and growth of blisters may be affected by the ion fluence [9].

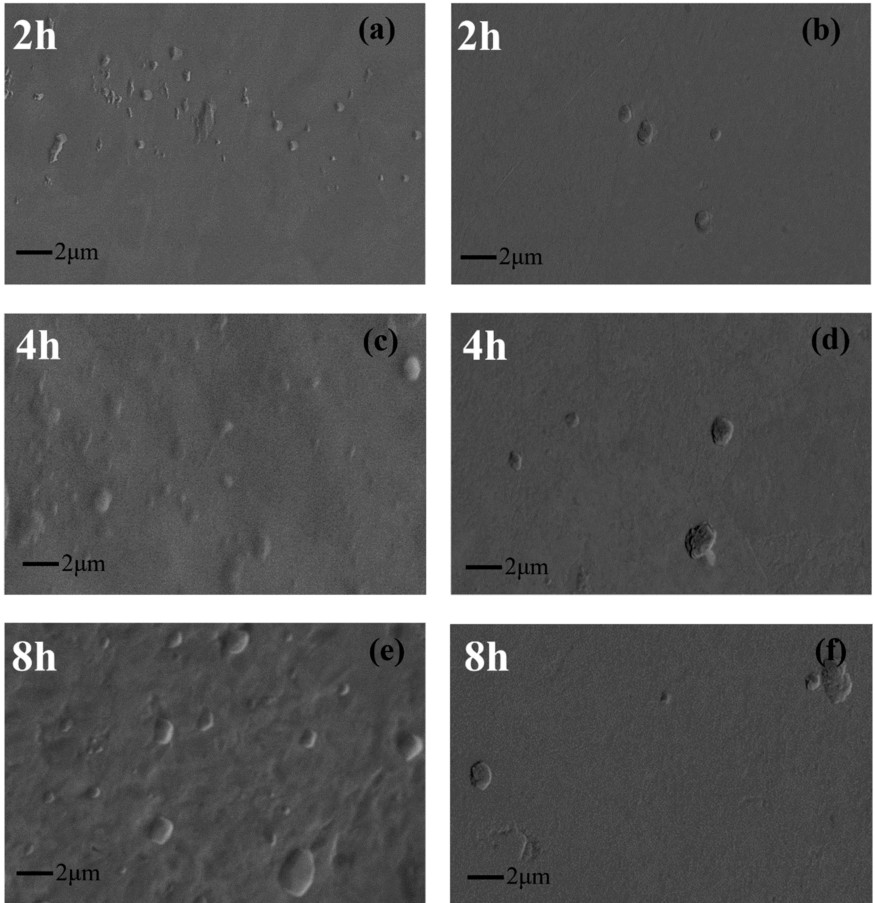

**Figure 4.** SEM images of the surfaces of W bulk (**a,c,e**) and nanocrystalline W films (**b,d,f**) exposed to 100 eV deuterium plasma under different fluences at 500 K: (**a,b**) irradiation for 2 h (fluence: $1.30 \times 10^{25}$ D/m$^2$); (**c,d**) irradiation for 4 h (fluence: $2.59 \times 10^{25}$ D/m$^2$); (**e,f**) irradiation for 8 h (fluence: $5.18 \times 10^{25}$ D/m$^2$).

*3.4. Cross-Section Imaging*

To better characterize the modifications of the top-most few micrometers below the surface after implantation, FIB is applied to the nanocrystalline W film surface and the cross-sectional morphology is observed by SEM. A protective layer of Pt is deposited onto the investigated surface in situ prior to the cross-sectioning. One selected blister is cut by FIB, and the cross-section image of the blister with a diameter of about 2 μm is shown in Figure 5, which clearly shows the formation of cracks below the surface. As shown in Figure 5, cracks are observed, extending deeply into the bulk. From the cross-sectional imaging, we can find that cracks extend down to 3 μm (Figure 5). As can be seen from Figure 5, the deepest crack depth is 3 μm while our film thickness is 6 μm. Therefore, the formation of cracks does not occur at the boundary between the deposit and the substrate nor within the substrate, but in the deposited layer. Similar cracks/voids were observed by Puschel et al. [10] and Jia et al. [11].

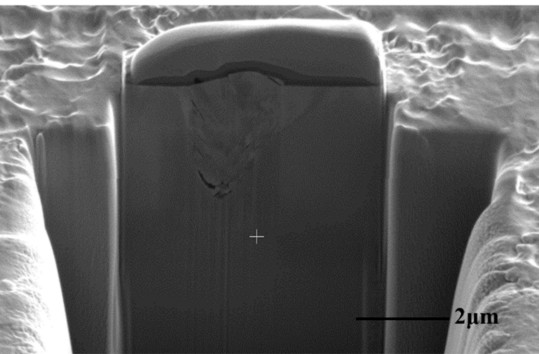

**Figure 5.** Focused ion beam (FIB) cross-section imaging of nanocrystalline W film after deuterium plasma irradiation.

### 3.5. Thermal Desorption Spectra

TDS is used to investigate the deuterium trapping information. Figure 6 shows the evolution of the TDS spectrum measured on W bulks and W films exposed to different fluences of deuterium plasma with ion energy of 100 eV at 500 K. The amount of deuterium desorption as a function of temperature for various implantation fluences is shown in Figure 6. To measure the deuterium retention, the samples are heated with a linear ramp of 10 K/min to a top temperature of 1273 K. Distinct desorption peaks correspond to distinct trap energies. In general, a high peak position corresponds to a high activation energy, and a low peak position is the opposite of a high peak position. Desorption from a deeper position of samples would result in the appearance of a high-temperature desorption peak.

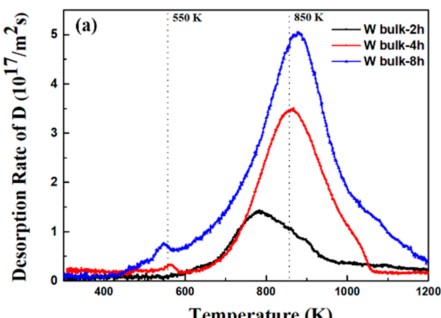 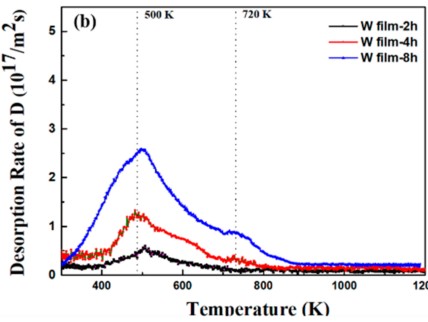

**Figure 6.** Thermal desorption spectroscopy (TDS) spectra of $D_2$ release in (**a**) W bulks and (**b**) W films exposed to 100 eV deuterium plasma at different irradiation fluences.

In Figure 6a, two kinds of desorption peaks of the W bulk can be distinguished, which are located at about 550 K and 850 K. The analysis of the TDS spectra of W films exposed to deuterium plasma shows the presence of peaks at 500 K and 720 K, as in Figure 6b.

The deuterium retention in the W film is $1.17 \times 10^{19}$ D/m$^2$ and in the W bulk is $2.23 \times 10^{19}$ D/m$^2$ exposed to the deuterium plasma with a fluence of $1.30 \times 10^{25}$ D/m$^2$ at 500 K. Increasing the fluence at the same exposure temperature results in a significant increase in deuterium retention, as shown in Figure 7.

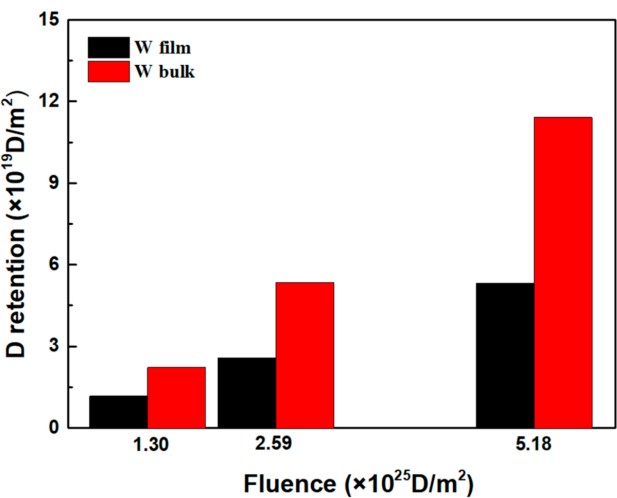

**Figure 7.** D retention in W bulks and W films at different exposure fluences.

## 4. Discussion

### 4.1. Hydrogen Plasma-Induced Blistering

Deuterium blister formation in tungsten samples may be induced by low-energy, high-flux deuterium irradiation [12]. The samples are immersed in the high-density deuterium plasma environment, and the deuterium atoms diffuse into W samples [13]. It has been reported that impurities, dislocations, vacancies, grain boundaries, and other crystal defects in the irradiation region are important trap sites for deuterium atoms. With regard to the formation of hydrogen isotope blisters, Liu et al. [14] reported a microscopic vacancy trapping mechanism based on first-principles calculations. Intrinsic defects such as vacancies in the tungsten material are the controlling factors for blister formation. Deuterium atoms prefer to be trapped by vacancies. Since these trap sites capture the deuterium atoms [13], the deuterium-vacancy clusters recombine and diffuse gradually into the tungsten material. It is reported that the concentration of deuterium in the near-surface layer during irradiation far exceeds that which the material can accommodate [15]. If the flux of implanted deuterium is greater than the rate at which deuterium diffuses out of the implanted area, a local supersaturation of deuterium appears [15]. The stress caused by local supersaturation of deuterium causes more deuterium atoms to reach the trap sites and recombine, leading to the establishment of a stress field.

In our experiments, with the irradiation fluence increasing, the blisters become increasingly obvious. The surface morphology changes indicate that the incident fluence has a large influence on the formation of the blisters. The ion energy (100 eV) is much lower than the sputtering and damage threshold of tungsten under deuterium bombardment [15]. Although Yang et al. demonstrated that there was no sputtering at a low energy of 100 eV, they found that blisters were formed using $100 \text{ eV}/10^{21} \text{ m}^{-2} \text{ s}^{-1}$ D plasma [16]. Blistering behavior occurs at the tungsten surface, even if the ion energy is too low to create displacement damage such as vacancies [15]. When more hydrogen atoms reach the lattice positions of tungsten, tungsten atoms are ejected from the lattice positions. Since the solubility of deuterium in tungsten is very low, the tungsten atoms diffuse to release the high stress, causing blistering [17]. Additional vacancies are created during this process [18], which trap more deuterium and provide extra conditions for deuterium accumulation. The hydrogen atoms aggregate and form blisters in tungsten [19].

Compared with those of W bulk, the size and density of the blisters on the surface of W film are smaller and lower, respectively, which may be due to the large volume fraction of grain boundaries of W film. The average grain size of W bulk is much larger than that of W film. The grain boundary volume fraction increases sharply if the grain size is smaller than 100 nm [20]. It has been reported that the high density of grain boundaries in W film enhances the threshold for hydrogen atoms to reach a supersaturated state, thus

inhibiting blister formation [21]. Therefore, hydrogen blister formation in W film is reduced in comparison with pure W bulk under the same conditions.

### 4.2. Sub-Surface Morphology Corresponding to Blisters

The cracks associated with the blisters appear beneath the surface, as shown in Figure 5. The stress caused by deuterium accumulation promotes the growth of the crack. Meanwhile, the material deforms and delaminates, causing blistering [22]. According to the results of Haasz et al. [23], there is a positive correlation between the blister diameter and the depth of the cracks. These blisters appear on the samples as the high fluence (long exposure time) allows the deuterium to accumulate at these large depths. It has been reported that many vacancies created by deuterium irradiation promote the formation of cracks far beyond the irradiation region, allowing deuterium to diffuse deeply into the sample, which is the possible reason for the deeper crack positions we observed [10].

### 4.3. Fluence Dependence of Deuterium Retention

Low temperature peaks of W films at around 500 K are observed in the TDS image in Figure 6b, which are caused by deuterium trapped in intrinsic defects related to dislocations, grain boundaries, and mono-vacancies [1,24,25]. According to the simulation, based on a trap energy of 0.8–1.2 eV, the presence of this deuterium peak position could be attributed to the trapping of deuterium at dislocations, grain boundaries, and single vacancies [26,27]. The high temperature peaks at 720 K are generally ascribed to the trapping from deuterium agglomerated in vacancy clusters [23,28,29]. As the fluence increases, the peaks become much higher, which indicates that deuterium diffuses to a deeper position.

The high temperature desorption peak of 720 K in the low-fluence ($1.30 \times 10^{25}$ D/m$^2$) W film is not obvious, whereas in the high-fluence ($5.18 \times 10^{25}$ D/m$^2$) W film this desorption peak is relatively clear. The appearance of the TDS peaks at 720 K is ascribed to a large number of vacancy-type defects generated by deuterium plasma irradiation [1]. In the case of low deuterium fluence ($1.30 \times 10^{25}$ D/m$^2$), only a few defects can be induced by deuterium plasma; this is not conducive for the formation of vacancy clusters, and thus the peak at around 720 K is not obvious. As the irradiation fluence increases, single vacancies aggregate to form clusters of vacancies. As can be seen in Figure 5b, with the increasing of the fluence up to $5.18 \times 10^{25}$ D/m$^2$, the peak at around 720 K transforms into a more obvious desorption peak, which confirms that the agglomeration of vacancies in clusters takes place when the fluence is increased. There are several reasons for the defects caused by deuterium plasma irradiation below the displacement threshold: plastic deformation by deuterium supersaturation [30] and the creation of vacancies [31]. The deuterium-induced stress that we mentioned above may cause single-vacancy movement and growth [32]. In addition, the duration of high-fluence irradiation in this experiment was 8 h; the longer duration allows the injected deuterium to diffuse into a larger depth, which is conducive to the diffusion of mono-vacancies to form vacancy clusters. The density of defects induced by plasma exposure increases with increasing duration and fluence.

The low temperature desorption peak of W bulk at 550 K exposed to low-fluence irradiation is not obvious, whereas in the high-fluence case this desorption peak is relatively clear (see Figure 6a). The low temperature peak at 550 K is attributed to the trapping of deuterium in the intrinsic defects. The W bulks (purity: 99.95%) used in this experiment were purchased commercially and have a higher relative density (99%) than the tungsten films (the relative density of 90%); thus, the intrinsic defects inside the W bulks are relatively few compared with the W films. With the increase in the fluence, the peaks at 550 K become obvious, which is due to the vacancies that are produced with the increase in fluence. Except for a small peak at 550 K, a main desorption peak appears at a higher temperature of 850 K. Vacancy defects created by low-energy deuterium are the main reason for the appearance of the high desorption peak at 850 K.

As compared to the low-fluence ($1.30 \times 10^{25}$ D/m$^2$) case, the intensity of each desorption peak becomes much higher for the high-fluence ($5.18 \times 10^{25}$ D/m$^2$) exposure, which is

considered to be due to the deuterium atoms diffusing into a deeper depth of material. For both the W bulks and the W films, there is a clear increase in overall retention as a function of plasma fluence/exposure time. The same deuterium release tendency is found, in which the deuterium retention rises with increasing fluences for W samples [24,32]. Irradiation studies by Shu et al. also show an increasing trend in deuterium retention with increasing deuterium fluence [33].

Surprisingly, the intensity of the low-temperature peak is much higher than that of the high-temperature peak in W film, whereas for the W bulk it is the opposite, which may be due to the large number of intrinsic defects in the film (relative density of 90%). The injected deuterium atoms prefer to fill up the intrinsic defects first. Compared with the W bulk, the W film has smaller grains and more grain boundaries. The stress can be uniformly dispersed in W film, while higher stress in the bulk drives more deuterium atoms to diffuse into the material. The deuterium-induced stress in the W bulk not only increases the movement of the vacancies but also facilitates deuterium diffusing into a larger depth [34], which leads to an increase in deuterium retention.

## 5. Conclusions

In this manuscript, the dependence of deuterium plasma fluence on surface morphology in W bulks and nanocrystalline W films was investigated. W bulks and W films were exposed to high-flux ($1.8 \times 10^{21}$ D/m$^2$s), low-energy (100 eV) deuterium plasma at 500 K. The blistering behavior in W bulks and W films was studied. The size of deuterium blisters is up to 2 μm after the high-fluence irradiation. Meanwhile, cracks are observed below the top surface. The formation of blisters and cracks may be ascribed to the nucleation points already existing in the material before plasma exposure. The formation of deuterium-vacancy clusters is considered to be the generic reason for blisters. Due to the low solubility of deuterium in the W lattice, high fluence exposure results in local supersaturation. The stress caused by the local supersaturation of deuterium causes more deuterium atoms to reach the trap sites and recombine. The increase in the size of blisters and corresponding cracks is probably due to the stress development in the material as the exposure time increases, prompting the diffusion of deuterium to a greater depth to reach more nucleation points.

The deuterium retention in W bulks and W films has been studied. Increasing deuterium fluence not only causes the mutation of the surface morphology but also increases the amount of trapped deuterium. As the fluence increases from $1.30 \times 10^{25}$ to $2.59 \times 10^{25}$ D/m$^2$, the amount of deuterium released into the nanostructure W films increased from $1.17 \times 10^{19}$ to $2.58 \times 10^{19}$ D/m$^2$. An increasing trend of the retention of deuterium with increasing incident deuterium fluence, up to the highest incident fluence used in this experiment ($5.18 \times 10^{25}$ D/m$^2$), is exhibited. The fluence dependence of deuterium retention in the W films is in accordance with the blistering behavior, which is caused by the high deuterium concentration in the irradiation area.

**Author Contributions:** Conceptualization, J.Y.; methodology, J.Y.; validation, J.Y.; investigation, J.Y.; writing—original draft preparation, J.Y.; writing—review and editing, X.L.; funding acquisition, K.Z. All authors have read and agreed to the published version of the manuscript.

**Funding:** This research was funded by the National Natural Science Foundation of China, grant number 51471015, 11675010, and 11775228.

**Institutional Review Board Statement:** Not applicable.

**Informed Consent Statement:** Not applicable.

**Data Availability Statement:** Not applicable.

**Conflicts of Interest:** The authors declare no conflict of interest.

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
