# Peer review of "Fluence Dependence of Surface Morphology and Deuterium Retention in W Bulks and Nanocrystalline W Films Exposed to Deuterium Plasma"

_applsci, doi:10.3390/app11041619_

Round 1
Reviewer 1 Report
I’m sorry to say that I don’t consider this an original piece of research, every observation and theory made is then followed with a reference to a paper that has already published that observation. I fail to see what the novelty is in this work. I thought the novelty may come from the nanocrystalline nature of the W compared with conventional W but this is never discussed. Also, there needs to be some quantitative analysis of the samples, it’s all observational and seems to be based off of just one sample.
The English standard is not of a high enough standard for publication and requires improving, there were multiple instances where I really struggled to understand what you were trying to say, please have it proof read.
I’d like to see more information about how you designed your experiments. Why did you use 100 eV, why 500K, why did you choose those particular fluences, how does that compare with what is expected in a fusion reactor.
Section 3.1, have you done any quantification, what was the areal density of blisters what was the average size etc. The images you present don’t really show me anything other than that there are blisters, maybe lower magnifications would be better to see more blisters.
Section 3.2 The FIB cross section doesn’t really show anything to me I can’t see multiple cracks and only a few voids, also which fluence is this sample, and what did the other fluences show? I don't really think this is very useful, maybe if you combined with some TEM there may be interesting observations.
Section 3.3 regarding the TDS the greatest release seems to be around 500K, but the irradiations were carried out at 500K so surely a huge amount of the D was being released during the irradiation, is that not the case?
Line 201-207, this is either a poor explanation of “trap-mutation/loop punching” or the authors are not clear on how this phenomenon takes place. A bubble of gas atoms can get to a sufficiently high pressure to displace atoms, but I’m not sure it’s likely they will reach the surface.
Figure 1 has a caption e) but no figure e)
Figure 2 I’m not sure what this figure is meant to be showing me, there is a label “blister” and an arrow that doesn’t seem to be pointing at anything. Is it the quality of the picture?
Figure 4 has captions of “Irratdiation”.
Reviewer 2 Report
The paper by Yan et al. reports on the characterization of the deuterium on polycrystalline tungsten films by magnetron sputtering; the blistering behavior and deuterium retention on tungsten films are assessed and correlated with the structure and composition of the films.
After reading this paper from the beginning to the end, I cannot find serious weakness for this manuscript. I have just three suggestions about the properties and research progress of these newly developed films. First, the revised manuscript should include the introduction and explanations for the advantages and future potential of the deuterium retention on tungsten films for semiconductor and biomedical devices. Then, structural identification of the deuterium retention on tungsten films should be further provided in the main text. Second, please ensure that the discussion of the possible interaction/repulsion mechanisms between deuterium and tungsten films is present in the main text. Finally, I strongly suggest that authors should point out what are contributions in this research paper and what are essentially different in the results from previously published articles.
Based on the comments above mentioned, this manuscript would be acceptable for Applied Sciences after major revision.
Reviewer 3 Report
The manuscript discusses the dependence of deuterium plasma fluence on the surface morphology and deuterium retention in nanocrystalline tungsten film deposited by dc magnetron sputtering. The deuterium exposure leads to formation of blisters and the size of the blisters depends on the fluence.
This is simple and well organized study.
The manuscript need some English corrections.
I have only a few minor comments:
i) line 78 - 79. "The specimens were annealed...." Does this refer to all the samples where a W film has been deposited.
ii) line 49 - 50 ".. by magnetron sputtering deposition,,," and line 128 should be magnetron sputtering deposited
iii) Line 93. Is the pressure up to 500000 Pa ? I do not think ECR can operate at this pressure.
iv) Line 100 it is stated that the sample temperture is kept at 500 K. Does this apply to the film surface.
v) line 156: "... observed by B. T. Puschel et al. [11] and Y.Z. Jia et al. [12]." should be "... observed by Puschel et al. [11] and Jia et al. [12]."
and line 238 "... by W.M. Shu et al. also show" should be "... by Shu et al. also show..."
vi) Line 89 - 90. There should be a better description of the ECR discharge and how the samples are exposed to the discharge and/or a reference to description of this system. How are the ions accelerated to 100 eV ?
Round 2
Reviewer 1 Report
While the authors have improved the paper, I do still feel that there aren’t any “new” observations in this work. Simply put the authors have irradiated W and nano-crystalline W and found that with sufficient dose blistering will occur and that more implantation leads to greater retention. As I said before the observations all have references showing that others have already reached these conclusions. I think if you did a study looking at the microstructures (maybe using TEM) of the different Ws and at the different fluences. Looking at how the Deuterium is trapped within the different samples, this would be a good paper.
Section 3.1 Why have you included this XRD data, what does it mean that there is a peak shift? What does it mean that the film peak shifts much more than the bulk. There is no discussion of this data.
Section 3.2 Same for the indentation work, why have you included it? What is the relevance? It’s not in the discussion at all.
Figure 4 is a great improvement, there is now an obvious difference between the fluences.
Section 3.3 cross section imaging. I’m still not sure what the significance of this is, with only one sample imaged I don’t know how that compares, and as you point out similar cracks/voids have already been shown in the literature.
Section 3.4. How long was it from the irradiation to the TDS measurement?
Section 4.3 “As the fluence increases, the peaks become much higher , which indicates that deuterium diffuses to a deeper position” does it indicate this, does it not just indicate that more deuterium was implanted thus there was more to be released?
Conclusion:
“The formation of deuterium vacancy clusters is considered as the generic reason for blisters.” How can you say this?
“increase of the size of blisters and corresponding cracks is probably due to the stress development in the material as the exposure time increases, prompting the diffusion of deuterium to a greater depth to reach more nucleation points.” The creation of blisters and large cavities is to accommodate the deuterium atoms, why would the lead to diffusion to a deeper depth?
Reviewer 2 Report
The authors have addressed my concerns and thus I recommend this manuscript for publication without further modification.